# Towards Linking Graph Topology to Model Performance for Biomedical Knowledge Graph Completion

**Alberto Cattaneo** [1]   **Thomas Martynec** [2]   **Stephen Bonner** [2]   **Carlo Luschi** [1]   **Daniel Justus** [1]

## Abstract

Knowledge Graph Completion has been increasingly adopted as a useful method for several tasks in biomedical research, like drug repurposing or drug-target identification. To that end, a variety of datasets and Knowledge Graph Embedding models has been proposed over the years. However, little is known about the properties that render a dataset useful for a given task and, even though theoretical properties of Knowledge Graph Embedding models are well understood, their practical utility in this field remains controversial. We conduct a comprehensive investigation into the topological properties of publicly available biomedical Knowledge Graphs and establish links to the accuracy observed in real-world applications. By releasing all model predictions and a new suite of analysis tools we invite the community to build upon our work and continue improving the understanding of these crucial applications.

## 1. Introduction and Background

Knowledge Graphs (KGs) have developed into an important tool to capture and represent knowledge within a domain, based on heterogeneous data originating from diverse sources. KGs represent facts in the form of triples $(h, r, t)$, connecting entities $h$ and $t$ via the relationship $r$. In biology-focussed KGs, entities correspond to genes, diseases, drugs or pathways (among others), which, together with their interactions, represent biological knowledge at a range of different abstraction levels. Knowledge Graph Embedding (KGE) models learn embeddings for all entities and relation types in a KG and can be used to infer likely but missing triples, a task known as Knowledge Graph

Completion (KGC). Since the adoption of representation learning for KGC, there has been an interest in linking the predictive performance of models to topological properties of KGs (Bordes et al., 2013). These properties have been largely divided into two categories: edge cardinalities and relational patterns (see Section 2). Early work considered the impact of relational patterns on validation metrics for different train/test splits (Toutanova & Chen, 2015). Believing that better ability to model these patterns is beneficial to predictive performance, it has become common practice to develop KGE models to specifically capture them (Wang et al., 2014; Yang et al., 2015; Sun et al., 2019; Chao et al., 2021; Yu et al., 2022). While such studies have provided valuable insights into the theoretical capabilities of KGE models, the prevalence of edge cardinalities and relational patterns in real-world datasets and their correlation with predictive performance has been less studied. A recent step in this direction investigates the frequency of graph topological patterns in public KGs from different domains (Teneva & Hruschka, 2023a;b), whilst other work explores their connection to the performance of KGE models (Ali et al., 2021; Jin et al., 2023). However, an obstacle arises in a lack of agreement regarding concrete definitions of several patterns across the literature. Further, prior studies do not focus on the biomedical domain, where KGC is increasingly used to support the complex and costly drug discovery process (Paliwal et al., 2020). Despite the fact that many biomedical KGs continue to be constructed (Königs et al., 2022; Chandak et al., 2023; Bonner et al., 2022a), it is still unclear how they topologically compare to general-domain ones, against which models are often developed. Also, quantifying the impact of differences in the graph structure on model predictions remains an open problem when it comes to complex tasks such as inferring new gene targets for a disease.

In this paper, we explore the relationship between biomedical KG topology and KGE model accuracy. Our main contributions are as follows:

- We provide an in-depth analysis of the topological properties of six public KGs, focussing on the biomedical domain, and compare the corresponding predictive performance of four well-established KGE models. While previous work has limited the investigation at

---

[1]Graphcore Research, Bristol, UK [2]Data Sciences and Quantitative Biology, Discovery Sciences, R&D, AstraZeneca, Cambridge, UK. Correspondence to: Alberto Cattaneo <albertoc@graphcore.ai>, Daniel Justus <danielj@graphcore.ai>.

*Accepted at the 1st Machine Learning for Life and Material Sciences Workshop at ICML 2024.* Copyright 2024 by the author(s).

the macro-level of relation types, we are able to detect stronger patterns linking topological properties to predictive accuracy by zooming in at the level of individual triples.

- We look in detail at highly relevant relations that biomedical practitioners are most interested in inferring. We give evidence of the topological differences in how they are represented in different KGs and how this reflects on predictive performance.

- To address inconsistencies in how KG properties are defined and utilized in the literature, we propose a standardized framework to describe KG topology and release a dedicated Python package `kg-topology-toolbox` implementing it[1].

Finally, to help bridge the gap between industry and academy, we make the predictions of the trained models available to the community (Cattaneo et al., 2024) to conduct further analysis.

## 2. Knowledge Graph Topological Properties

For a triple $(h, r, t)$ in a KG $\mathcal{G}$ we define the *head out-degree* as the cardinality of the set $\{\hat{t} \mid \exists \hat{r} : (h, \hat{r}, \hat{t}) \in \mathcal{G}\}$ and, analogously, the *tail in-degree* as the cardinality of $\{\hat{h} \mid \exists \hat{r} : (\hat{h}, \hat{r}, t) \in \mathcal{G}\}$. We further define the *head out-degree of same relation* $\deg_r(h)$ and the *tail in-degree of same relation* $\deg_r(t)$ as the cardinalities of $\{\hat{t} \mid (h, r, \hat{t}) \in \mathcal{G}\}$ and $\{\hat{h} \mid (\hat{h}, r, t) \in \mathcal{G}\}$, respectively. The *edge cardinality* of a triple $(h, r, t)$ is then defined as follows:

| $\deg_r(h)$ \ $\deg_r(t)$ | = 1 | > 1 |
|---|---|---|
| = 1 | *one-to-one* | *many-to-one* |
| > 1 | *one-to-many* | *many-to-many* |

We also define the following *edge topological patterns* for triples $(h, r, t) \in \mathcal{G}$:

- $(h, r, t)$ *is symmetric* $\iff h \neq t$ and $(t, r, h) \in \mathcal{G}$;
- $(h, r, t)$ *has inference* $\iff \exists r' \neq r : (h, r', t) \in \mathcal{G}$;
- $(h, r, t)$ *has inverse* $\iff \exists r' \neq r : (t, r', h) \in \mathcal{G}$;
- $(h, r, t)$ *has composition* $\iff \exists r_1, r_2$ and $n \notin \{h, t\} : (h, r_1, n), (n, r_2, t) \in \mathcal{G}$.

In the biomedical domain, a compositional pattern could, for instance, be useful to infer that a drug can treat a disease because they share a common gene connection. Similarly, an inference pattern might be used to infer potentially novel

therapeutic gene-disease connections by aggregating multiple types of gene-disease associations. See Figure A.1 for an illustration of edge cardinalities and edge topological patterns and additional notation. It is important to note that a triple can satisfy multiple topological patterns simultaneously, or none at all.

While the above cardinalities and topological patterns have an intuitive definition for individual triples, previous studies (Jin et al., 2023; Teneva & Hruschka, 2023a) have considered them as properties of relation types as a whole, despite the fact that a given property might be satisfied only by a fraction of the triples of a certain relation. We find that such conceptual generalization lacks a proper formalization in the literature, with inconsistencies in how it is performed, and is prone to introducing noise in results (see Section 4.1). Thus, we consider the above defined properties only as properties of triples.

## 3. Experimental Setup

We investigate five public biomedical KGs: Hetionet (Himmelstein et al., 2017), PrimeKG[2] (Chandak et al., 2023), PharmKG (Zheng et al., 2021), OpenBioLink (Breit et al., 2020), PharMeBINet (Königs et al., 2022) (detailed in Table A.1). We also include the trivia KG FB15k-237 (Toutanova & Chen, 2015) as a baseline, to detect any results unique to the biomedical domain. On all KGs we train four of the most popular KGE models: TransE (Bordes et al., 2013), DistMult (Yang et al., 2015), RotatE (Sun et al., 2019) and TripleRE (Yu et al., 2022), which, whilst being all well-established among practitioners (Hu et al., 2020), differ in complexity and in the relational patterns they can capture (Table A.3). In particular, TransE and DistMult model the interaction between head entity, relation type and tail entity in quite simple terms. Nevertheless, despite missing certain topological patterns, they remain strong baselines. On the other hand, RotatE and TripleRE are more recent approaches able to capture all four investigated topological patterns. Notice however that the theoretical capability of a scoring function to model a particular edge topological pattern is not per se a guarantee of stronger predictive performance on such edges (Jin et al., 2023). Our experiments are designed to quantify this impact.

The training scheme and hyperparameter optimisation details are presented in Appendix B. The results reported in the following sections refer to tail predictions generated on the held out test split, by scoring each $(h, r, ?)$ query against all entities in the KG and computing the rank of the ground truth tail $t$, after masking out scores of other $(h, r, t')$ triples contained in the graph.

---

[2]In the original PrimeKG graph, for every triple $(h, r, t)$ the reverse triple $(t, r, h)$ is present as well. We prepocessed PrimeKG to remove these reverse edges.

# 4. Results

## 4.1. Effect of Topological Properties on MRR

We observe a significant variance in mean reciprocal rank (MRR) across the different KGs and KGE models (Figure 1). In an attempt to understand its cause, we focus our analysis on the edge cardinalities and topological patterns described in Section 2. We find that they occur with varying frequencies in the different KGs (Figure A.2 and Table A.2). However, the data does not support a conclusion about the effect of these topological properties on model accuracy when only considering their average occurrence per dataset (Figure C.1). Previous work (e.g., Teneva & Hruschka (2023a)) went one step further, classifying relation types based on the predominant edge topological pattern/cardinality, however this also does not yield conclusive results (Figure C.2). This is due to the fact that the confounding effects of covariates, such as node degrees and different topological patterns, remain too difficult to disentangle, as these properties are often not homogenous within a relation type. Therefore, we dissect the link between topological properties and KGE models accuracy on the level of individual triples to allow for a finer-grained analysis with improved statistical power.

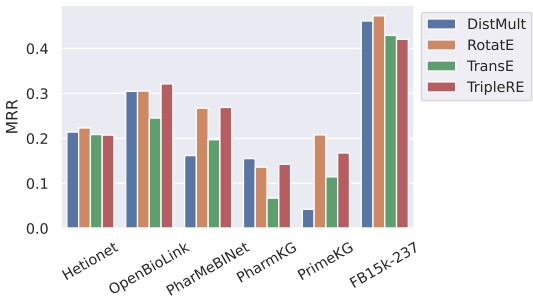

Figure 1: Mean reciprocal rank on the test split achieved by different models, for the six datasets.

While all investigated KGs are dominated by many-to-many triples (Figure A.2), there is a wide variability in the effect of edge cardinality on MRR across different KGs (Figure 2). This suggests that the exact entity degrees, together with other topological properties, might be better suited to explain the MRR than a binary one/many cardinality classification. Indeed, for a given triple, we observe a strong correlation between model accuracy and both the out-degree of the head entity and the in-degree of the tail entity (Figure C.3), especially for degrees of same relation (Figures 3 and C.4; see Figure A.3 for the distribution of degrees in the different KGs). In fact, a high in-degree of the tail node in a tail prediction task has been linked to a higher score (Bonner et al., 2022b), therefore increasing the likelihood of predicting it, as confirmed in Figure C.5. On the other hand, a high out-degree of same relation of the head node in a

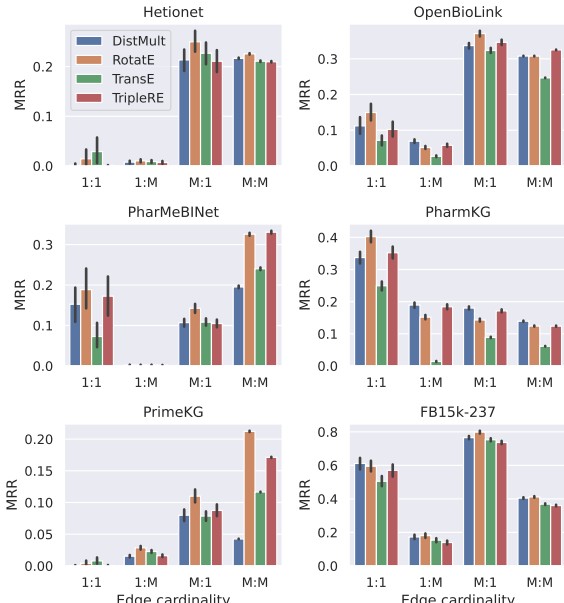

Figure 2: Effect of edge cardinality on MRR.

query $(h, r, ?)$ implies multiple correct tail entities. Some of these might not be present in the graph and thus cannot be filtered out during inference, making the task of predicting the specific entity that is expected by the model harder.

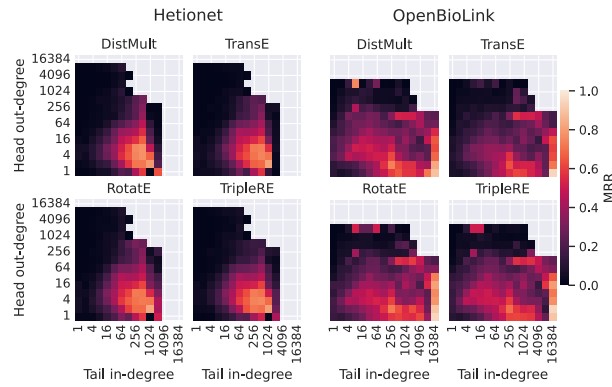

Figure 3: Effect of the head and tail degrees of same relation type on MRR.

Consequently, to reduce confounding, we investigate the effect of edge topological patterns within sets of triples with similar head and tail degrees. We find that such effects are more relevant if the degrees of head and tail entity are small. When this is the case, compositions are beneficial to MRR across all datasets and models (Figure 4 and Figure C.6). Interestingly, this is true even for DistMult that can't explicitly model compositional patterns.

For the patterns *is symmetric*, *has inference* and *has inverse*

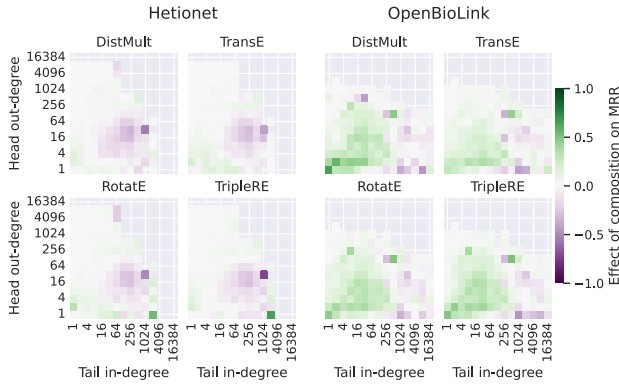

Figure 4: The effect of having compositions on MRR, triples grouped by their head and tail degrees of same relation.

we need to distinguish whether the counterpart of the tested edge was present in the training split. In this case, the prediction typically gets much easier due to the availability of additional information (Toutanova & Chen, 2015), see Figure C.7. An exception to this rule is the prediction of symmetric triples with the scoring function TransE. This is unsurprising since TransE by design cannot model symmetric relations – a factor likely contributing to the poor performance of TransE on OpenBioLink and PharmKG, which have a large fraction of symmetric triples (Figure 1 and Table A.2). On the other hand, our experiments confirm an above average performance of DistMult (which models all relations as symmetric) on symmetric triples (Figure C.7). In the more challenging case where the counterpart edge is not present in the training data, symmetry shows a detrimental effect on MRR whereas having inverse or inference has only little impact on accuracy (Figure C.7).

Note that all the above remarks apply equally to biomedical KGs and the general-domain KG FB15k-237.

## 4.2. Predicting Specific Relation Types

To apply the analysis conducted in the previous section to real-world tasks, we focus on the relations between pairs of entity types that are of particular interest to practitioners. We investigate how they are represented in different biomedical KGs, and how these differences affect the predictive performance of KGE models. Statistics for the interactions of the considered entity types are given in Figure C.8, while MRR is plotted in Figure 5.

As displayed in Figure C.9, the trained models tend to consistently predict entities of the correct type (for instance, gene entities when predicting drug-gene interactions), albeit with different levels of accuracy. Interestingly, DistMult often shows worse capabilities in this so-called demixing task than other scoring functions. As a consequence of demixing,

the size of the potential set of tails for a given interaction needs to be taken into consideration when comparing MRR across different datasets and relation types (the smaller the set of candidates, the higher the expected MRR achieved by random ranking).

**Gene-Gene**. As shown in Figure C.8a, these interactions are characterized by a large number of potential tails and a large head out-degree of same relation (with the exception of PharmKG for the latter), making predictions hard. However, a high MRR is observed for PharmKG and OpenBioLink: this is explained by the fact that, for these datasets only (Figure C.8b), most triples are symmetric (and a significant portion also have inverse/inference edges), with the counterpart likely to have been seen during training (Section 4.1). This also explains the relative ordering of scoring functions for OpenBioLink and PharmKG, with DistMult being the best and TransE the worst.

**Drug-Gene**. In Hetionet, PrimeKG and OpenBioLink these interactions are easier to predict than gene-gene interactions (Figure 5), likely due to an overall smaller number of potential tails seen during training and, in the case of PrimeKG, a markedly smaller head out-degree. PharmKG and PharMeBINet do not satisfy either of these properties, and indeed we observe no improvements in MRR. The strong predictive power observed for OpenBioLink can be linked to the presence of inverse edges for the vast majority of these triples.

**Drug-Disease**. Despite a small number of training triples (which is reflected in the sub-optimal demixing profile, Figure C.9c), drug-disease interactions are easily predicted in Hetionet as only 91 disease entities appear as tails of such triples. Remarkably good performance across all scoring functions is observed for PrimeKG, where the number of candidate tails is larger but the out-degree of head entities remains contained. When both these parameters increase (as in PharMeBINet and PharmKG) performance visibly degrades, despite larger in-degree of tails.

## 4.3. Case Study: Effect of Additional Training Data

Even when focussing on specific interactions that are represented in multiple KGs, a direct comparison of KGE predictive power across different datasets is limited by having to compare different test sets. In the case of Hetionet and PharMeBINet, however, the latter is constructed by augmenting the former with data coming from other biological databases (Königs et al., 2022). One can therefore often match individual triples in Hetionet to their exact counterpart in the larger PharMeBINet, which opens up the possibility of studying at a more fundamental level the impact of additional training data on the predictive performance of KGE models. This is relevant to practitioners as they construct the training KG for a specific task, which is usu-

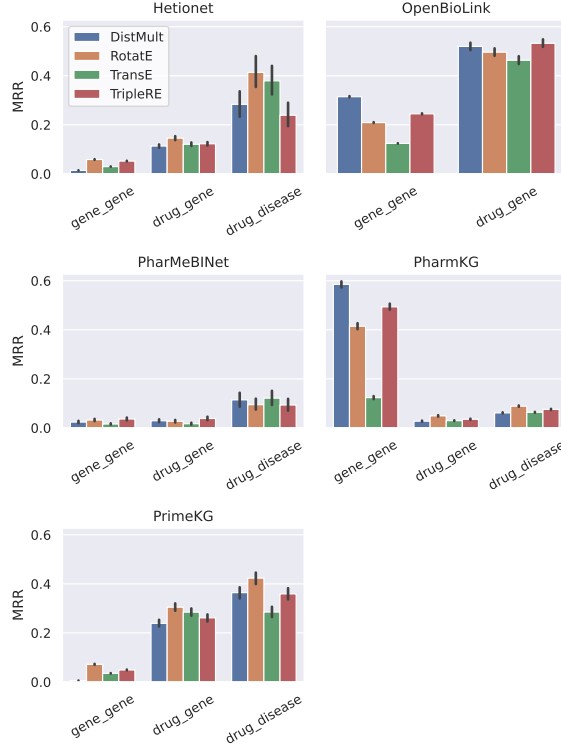

Figure 5: MRR of specific interaction types.

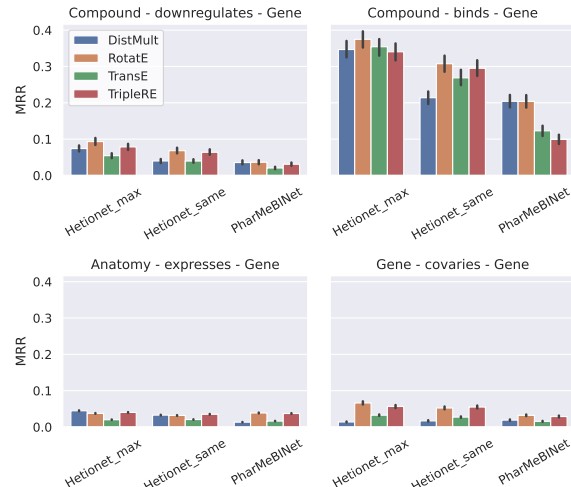

Figure 6: Comparison of models trained on Hetionet and PharMeBINet when testing MRR on a set of common edges.

ally done by sub-sampling relevant triples from larger (often proprietary) databases (Chandak et al., 2023).

We consider eight relation types where we can find a significant number of common triples between the two datasets (statistics are given in Table C.1). We extract 10% of the shared triples of the considered relation type as test set and use all other edges in each graph for training. In the case of Hetionet, in addition to the embedding size selected in Section 4.1 maximizing memory utilization, we also train models with the same embedding size chosen for PharMeBI-Net, which is generally strictly smaller (*Hetionet_max* and *Hetionet_same* in Figure 6; hyperparameters in Table B.1). To further ensure a fair comparison, at test time we restrict predictions to a custom set of candidate tails that is the same for the two datasets, namely the set of entities appearing as tails in shared triples of the given relation type.

As shown in Figures 6 and C.10, across all tested relation types there is no indication that the KGE models are able to benefit from the additional data seen when training on the larger PharMeBINet graph. On the contrary, models trained on Hetionet consistently perform better, even when using the same embedding size. Interestingly, the gap in MRR varies strongly across relation types and also across scoring functions, with DistMult showing generally little difference when comparing models trained with the same

embedding size, while distance-based scoring functions experience worse degradation. From Table C.1 we notice that the relation types where the MRR gap is more significant tend to have a larger overall head out-degree (and more relation types coming out from head nodes) for triples in PharMeBINet, compared to Hetionet. We hypothesise that the fact that these nodes are used as head entities for more relation types and triples, many of which are likely not relevant for the specific task at hand, negatively impacts the quality of embeddings. On the other hand, *Compound-downregulates-Gene* and *Compound-causes-Side Effect* exemplify relation types where PharMeBINet contains far more triples in addition to the ones in Hetionet. Even in this case, where we could expect the additional training data in the larger dataset to be strictly relevant to the prediction task, all scoring functions show markedly degraded performance when trained on PharMeBINet.

All this suggests that, in scenarios where the memory budget is fixed, training on smaller, tailored graphs and increasing the embedding size could be more beneficial than expanding the size of the KG, as the additional data can be a source of confusion for shallow KGE models.

## 5. Conclusions

This paper analyses the topological properties of widely used biomedical KGs and compares the corresponding predictive performance of different KGE models, focussing in particular on link-prediction tasks relevant to practitioners. Deviating from previous studies, we justify the need to look at properties of individual edges when trying to explain results, as pooling at the relation level introduces too much noise to detect any patterns. By going beyond

the coarse one/many binary classification typically used for edge cardinality, we find that considering the actual degrees of head and tail nodes gives a stronger predictor explaining model accuracy. Interestingly, similar interpretations apply to both biomedical KGs and the general-domain KG. Edge topological patterns also impact predictive accuracy, especially when entity degrees are small. For such patterns, we observe an improved accuracy when the counterpart edge (e.g., the reverse edge for symmetric triples) has been seen during training. This in turn raises the problem of inconsistent model validation schemes and topological property definitions when comparing results of previous studies on similar datasets. We address this problem by releasing all predictions from our experiments, together with a new toolbox for KG topological analysis. Finally, by performing a case study comparing predictions on common sets of edges shared by different KGs, we show that training on larger graphs, encoding more biomedical data, can unintuitively harm predictive performance. This evidence should encourage a wider discussion on the guiding principles to adopt when constructing the training KG for biomedical tasks – a crucial problem for real-world practitioners, but scarcely investigated in the literature.

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

## A. Edge, Dataset and Model Properties

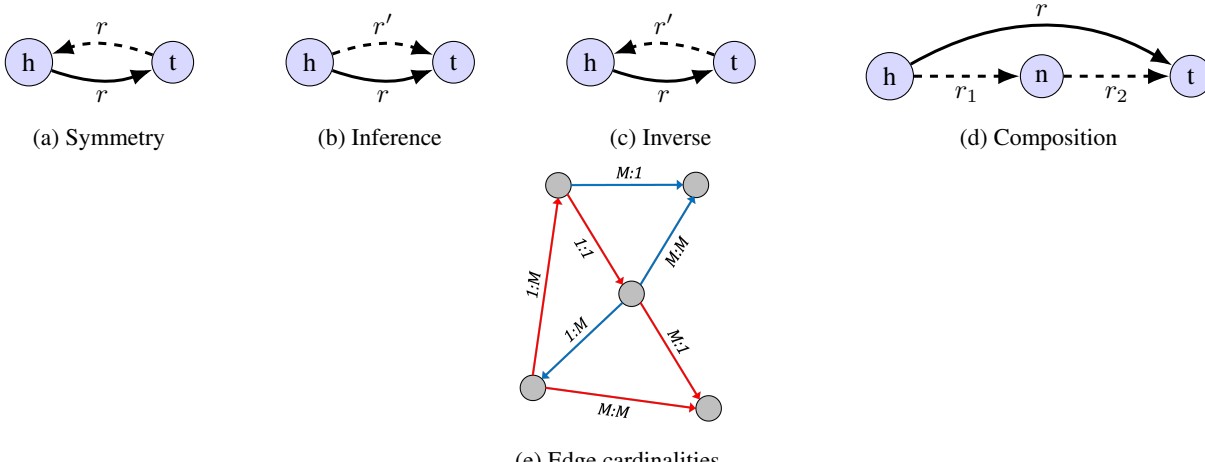

(a) Symmetry     (b) Inference     (c) Inverse     (d) Composition

(e) Edge cardinalities

Figure A.1: (a-d) Edge topological patterns. We refer to the dashed edge(s) as *counterpart edge* of $(h, r, t)$. (e) Example of edge cardinalities in a KG with two relation types (blue, red). These are defined using the head/tail degree of same relation.

Table A.1: Dataset properties.

| Graph | # Entities | # Relations | # Triples | Avg node degree |
|---|---|---|---|---|
| Hetionet | 45,158 | 24 | 2,250,197 | 99.66 |
| OpenBioLink | 184,635 | 28 | 4,563,405 | 49.43 |
| PharMeBINet | 2,653,751 | 208 | 15,883,653 | 11.97 |
| PharmKG | 188,296 | 39 | 1,093,236 | 11.61 |
| PrimeKG | 129,375 | 30 | 4,050,064 | 62.61 |
| FB15k-237 | 14,541 | 237 | 310,116 | 42.65 |

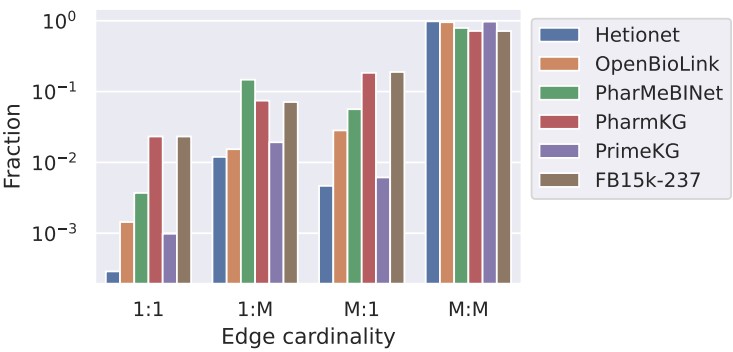

Figure A.2: Occurrence of edge cardinalities in the datasets.

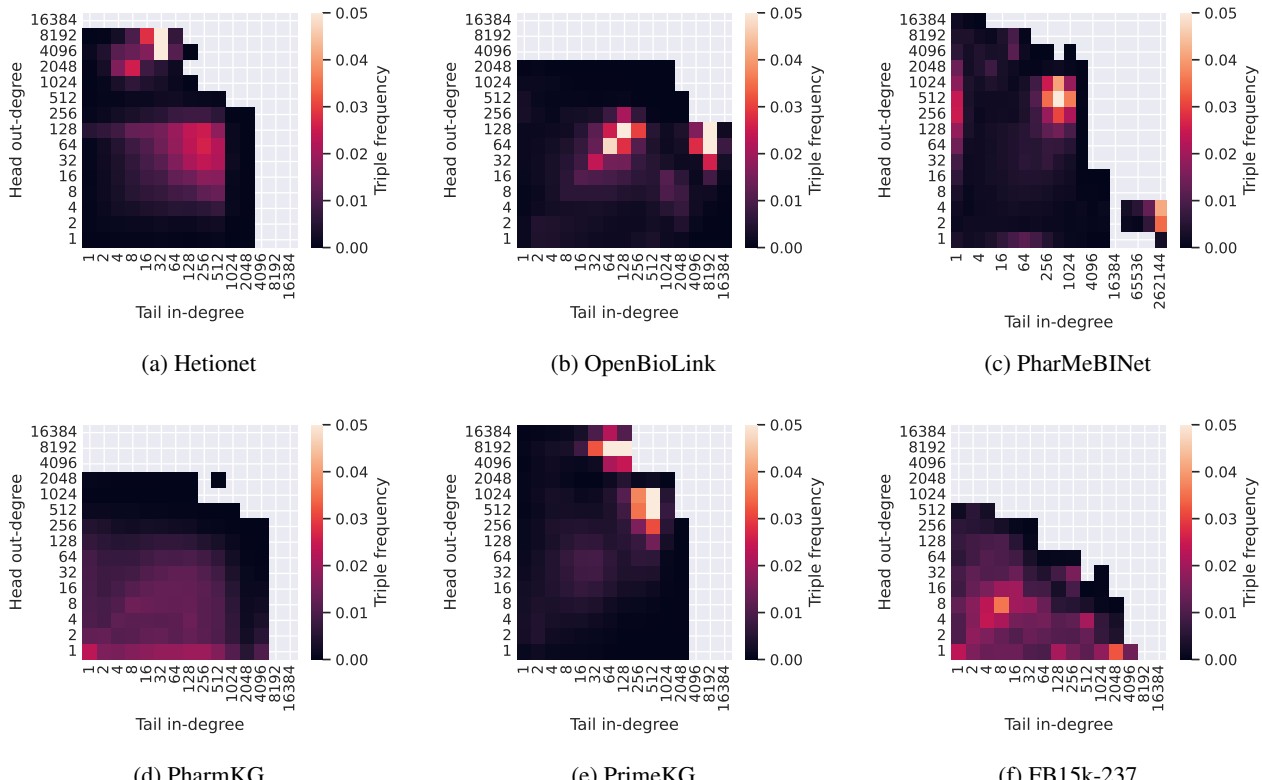

Figure A.3: Relative frequency of triples when grouped by head out-degree and tail in-degree of the same relation type.

Table A.2: Occurrence of edge topological patterns as fraction of total triples in the datasets. Note that, since PrimeKG was pre-processed to remove reverse edges as described in Section 3, it does not contain any symmetric triples.

| Graph | Symmetry | Inference | Inverse | Composition |
|-------|----------|-----------|---------|-------------|
| Hetionet | 0.002 | 0.124 | 0.001 | 0.693 |
| OpenBioLink | 0.317 | 0.372 | 0.359 | 0.840 |
| PharMeBINet | $2.420 \times 10^{-4}$ | 0.052 | 0.002 | 0.598 |
| PharmKG | 0.197 | 0.124 | 0.059 | 0.651 |
| PrimeKG | 0 | $2.081 \times 10^{-4}$ | 0 | 0.807 |
| FB15k-237 | 0.113 | 0.161 | 0.217 | 0.645 |

Table A.3: Scoring functions and their ability to model four fundamental relation properties: S = Symmetry; INF = Inference; INV = Inversion; C = Composition. For RotatE we assume $d$ even and denote by $\mathbb{C}^{\frac{d}{2}}$ the vector space $\mathbb{R}^d = (\mathbb{R} \oplus i\mathbb{R})^{\frac{d}{2}}$ with the structure of $\mathbb{R}$-algebra induced by the product of complex numbers. $\circ$ denotes the Hadamard product; $p \in \{1, 2\}$.

| Model | Scoring function | | S | INF | INV | C |
|-------|------------------|--|---|-----|-----|---|
| TransE | $-\|\boldsymbol{h} + \boldsymbol{r} - \boldsymbol{t}\|_p$ | $\boldsymbol{h}, \boldsymbol{r}, \boldsymbol{t} \in \mathbb{R}^d$ | ✗ | ✓ | ✓ | ✓ |
| RotatE | $-\|\boldsymbol{h} \circ e^{i\boldsymbol{r}} - \boldsymbol{t}\|_p$ | $\boldsymbol{h}, \boldsymbol{t} \in \mathbb{C}^{\frac{d}{2}}, \boldsymbol{r} \in \mathbb{R}^{\frac{d}{2}}$ | ✓ | ✓ | ✓ | ✓ |
| DistMult | $\langle \boldsymbol{r}, \boldsymbol{h}, \boldsymbol{t} \rangle$ | $\boldsymbol{h}, \boldsymbol{r}, \boldsymbol{t} \in \mathbb{R}^d$ | ✓ | ✓ | ✗ | ✗ |
| TripleRE | $-\|\boldsymbol{h} \circ \boldsymbol{r_h} + \boldsymbol{r_m} - \boldsymbol{t} \circ \boldsymbol{r_t}\|$ | $\boldsymbol{h}, \boldsymbol{r_h}, \boldsymbol{r_m}, \boldsymbol{t}, \boldsymbol{r_t} \in \mathbb{R}^d$ | ✓ | ✓ | ✓ | ✓ |

# B. Details on Experimental Setup and Hyperparameter Selection

All datasets were randomly split into training, validation and test set (80% / 10% / 10%; in the case of PharMeBINet, 99.3% / 0.35% / 0.35% to mitigate the increased inference cost on the larger dataset). To ensure comparability across KGs, this random split was used even if pre-defined training, validation and test sets were provided with a dataset. We adopted log-sigmoid loss with negative adversarial sampling (Sun et al., 2019) and margin 12.0, and the Adam optimiser (Kingma & Ba, 2015) for updating parameters. During training we always used negative sample sharing (Cattaneo et al., 2022). All experiments were performed on Graphcore IPUs using the BESS framework[3] (Cattaneo et al., 2022). A fixed batch size of 128 triples per device (192 for PharMeBINet, with gradient accumulation) was adopted, while the embedding size for entities and relations was chosen for each KG and each scoring function independently to maximise the memory utilisation of a Bow-2000 IPU machine with 4 IPU processors (in the case of PharMeBINet, a Bow Pod$_{16}$ with 16 IPUs). This is to ensure a fair comparison between scoring functions with different memory costs. For some scoring functions, especially DistMult, the memory footprint is typically dominated by the model parameters, allowing a larger hidden size for smaller KGs. For other scoring functions, especially TripleRE, memory is typically dominated by activations, resulting in a similar hidden size for differently sized KGs. The learning rate, the norm used by the scoring function (L1 or L2) and the number of negative samples were determined by a hyperparameter sweep, based on the validation MRR (Table B.1).

Table B.1: Experiment hyperparameters for different datasets. *Hetionet_same* refers to the alternative experimental configuration used in Section 4.3.

| Graph | Model | Hidden size | Learning Rate | Scoring Norm | # Negative samples / positive |
|---|---|---|---|---|---|
| Hetionet | DistMult | 2048 | 0.0003 | - | 16 |
| | RotatE | 512 | 0.001 | L2 | 16 |
| | TransE | 1024 | 0.0001 | L1 | 16 |
| | TripleRE | 384 | 0.0001 | L1 | 16 |
| Hetionet_same | DistMult | 300 | 0.0003 | - | 16 |
| | RotatE | 128 | 0.003 | L2 | 16 |
| | TransE | 256 | 0.0003 | L1 | 16 |
| | TripleRE | 256 | 0.0001 | L1 | 16 |
| OpenBioLink | DistMult | 768 | 0.0003 | - | 16 |
| | RotatE | 256 | 0.003 | L2 | 16 |
| | TransE | 512 | 0.0001 | L1 | 16 |
| | TripleRE | 256 | 0.001 | L2 | 16 |
| PharMeBINet | DistMult | 300 | 0.003 | - | 16 |
| | RotatE | 128 | 0.001 | L2 | 16 |
| | TransE | 256 | 0.00003 | L1 | 16 |
| | TripleRE | 256 | 0.0001 | L2 | 16 |
| PharmKG | DistMult | 768 | 0.003 | - | 16 |
| | RotatE | 384 | 0.003 | L2 | 16 |
| | TransE | 768 | 0.0001 | L1 | 16 |
| | TripleRE | 384 | 0.0003 | L1 | 16 |
| PrimeKG | DistMult | 1024 | 0.0003 | - | 16 |
| | RotatE | 384 | 0.001 | L2 | 16 |
| | TransE | 768 | 0.0001 | L1 | 16 |
| | TripleRE | 256 | 0.0001 | L1 | 16 |
| FB15k-237 | DistMult | 4096 | 0.001 | - | 16 |
| | RotatE | 1024 | 0.003 | L2 | 16 |
| | TransE | 2048 | 0.0001 | L1 | 16 |
| | TripleRE | 256 | 0.001 | L1 | 16 |

---

[3]https://github.com/graphcore-research/bess-kge

# C. Additional Results

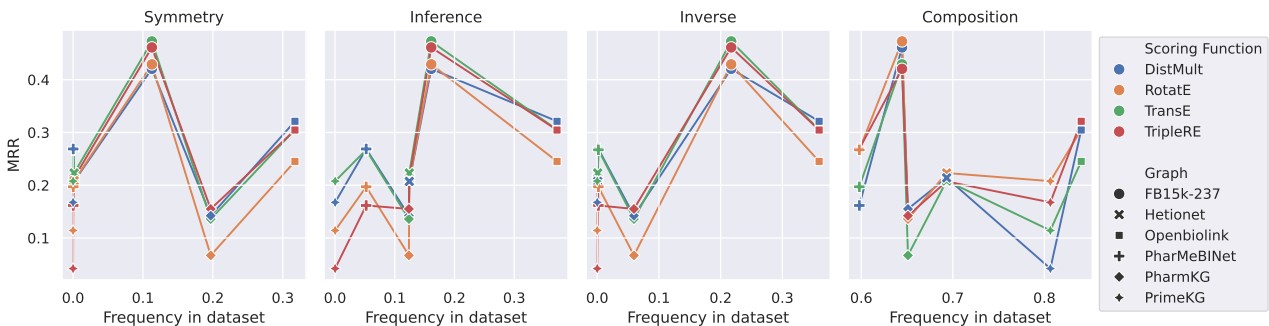

Figure C.1: Mean reciprocal rank plotted against the average occurrence of edge topological patterns in the six datasets.

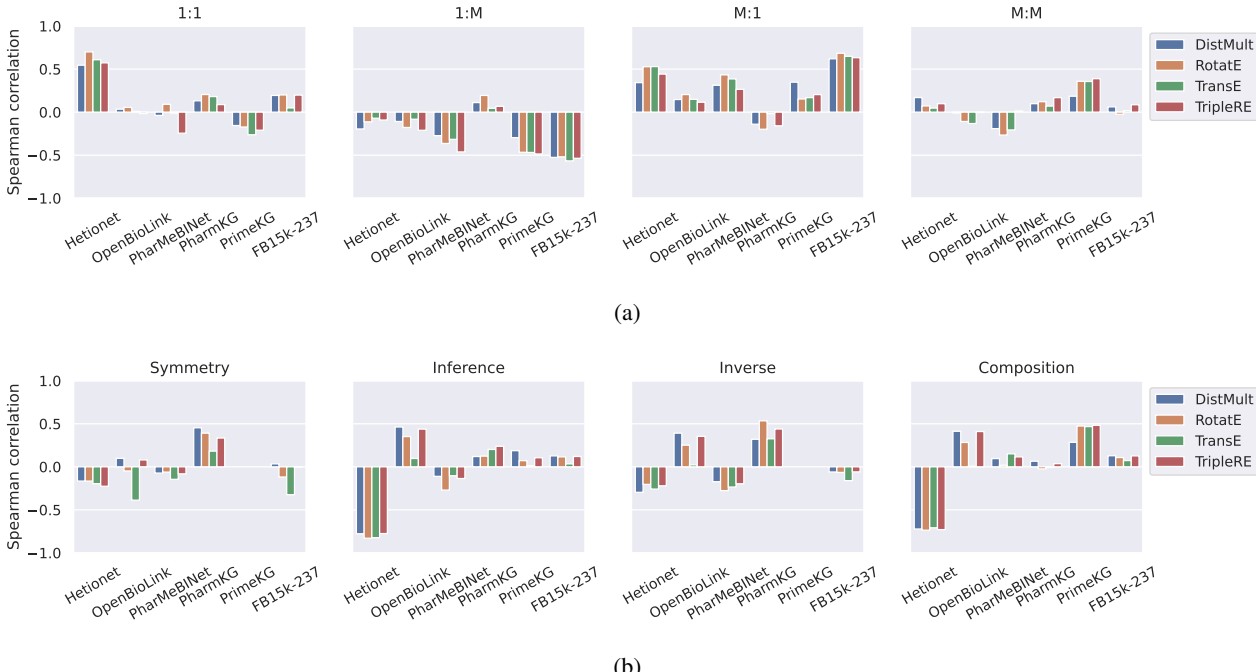

Figure C.2: Spearman-rank correlation between the average MRR of a relation type and the average frequency of edge cardinalities (a) and of topological patterns (b) in that relation type.

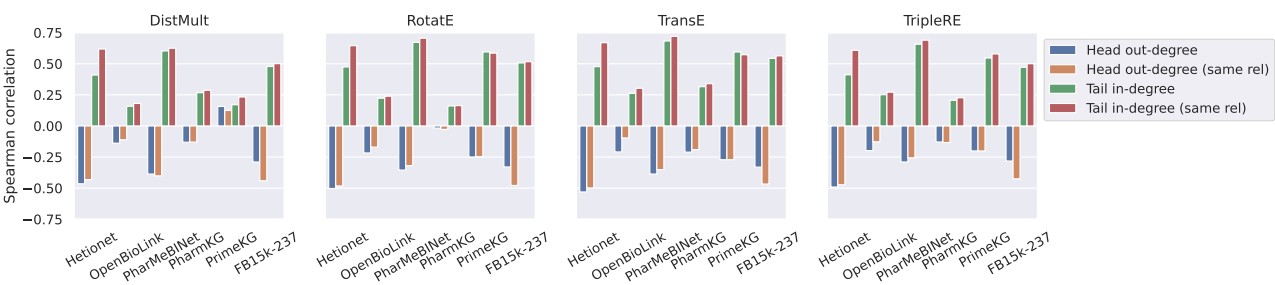

Figure C.3: Spearman-rank correlation between MRR of individual triples and the out-degree of the head node as well as in-degree of the tail node.

(a) PharMeBINet

(b) PharmKG

(c) PrimeKG

(d) FB15k-237

Figure C.4: Effect of head out-degree and tail in-degree on MRR for additional datasets.

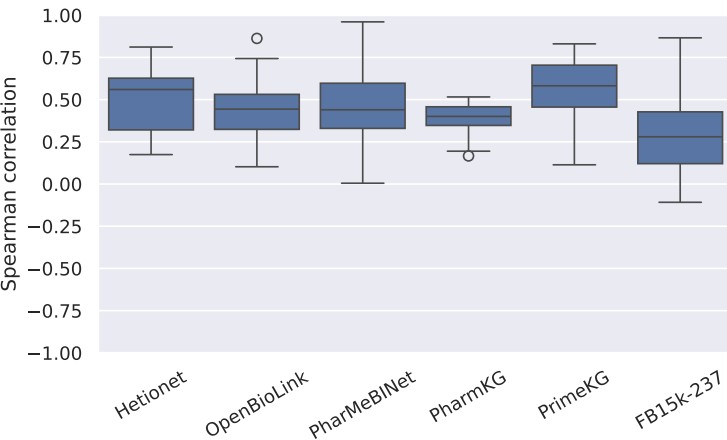

Figure C.5: Distribution of Spearman-rank correlation between in-degree of same relation and how frequently the entity is *incorrectly* selected among the top-100 tail predictions, grouping test queries by relation type. A positive correlation means that KGE models are biased towards predicting entities with a larger number of incoming edges of the relation type considered in the query.

Table C.1: Statistics of the relation types compared between Hetionet and PharMeBINet in Section 4.3. *Matching rate* denotes the fraction of triples of the relation type that are matched to triples in the other dataset. For the rows *Unique heads/tails/out-relations/in-relations* and *Head/Tail out-/in-degree (same relation)*, the median value is reported. For the rows *Has inverse/inference/composition*, we report the fraction of total relation triples with the given property.

| Relation | Disease-localizes-Anatomy | | Compound-binds-Gene | | Gene-covaries-Gene | | Anatomy-expresses-Gene | |
|---|---|---|---|---|---|---|---|---|
| Dataset | Hetionet | PhMBNet | Hetionet | PhMBNet | Hetionet | PhMBNet | Hetionet | PhMBNet |
| Relation triples | 3602 | 3602 | 11571 | 11622 | 61690 | 61615 | 526407 | 526180 |
| Matching rate | 1.0 | 1.0 | 0.998 | 0.993 | 0.999 | 1.0 | 0.999 | 1.0 |
| Test triples | 360 | 360 | 1141 | 1141 | 6161 | 6161 | 52618 | 52618 |
| Unique heads | 133 | 133 | 1389 | 1426 | 9043 | 9034 | 241 | 241 |
| Unique tails | 398 | 398 | 1689 | 1701 | 9542 | 9518 | 18094 | 18074 |
| Head out-degree | 212 | 227 | 132 | 2085 | 74 | 128 | 11952 | 11945 |
| Head out-degree s.r. | 34 | 34 | 14 | 14 | 20 | 20 | 7937 | 7935 |
| Unique out-relations | 4 | 6 | 4 | 20 | 4 | 8 | 3 | 3 |
| Tail in-degree | 11 | 11 | 102 | 162 | 83 | 114 | 77 | 112 |
| Tail in-degree s.r. | 11 | 11 | 36 | 36 | 17 | 17 | 44 | 44 |
| Unique in-relations | 1 | 1 | 7 | 13 | 6 | 11 | 6 | 11 |
| Has inverse | 0.0 | 0.0 | 0.0 | 0.03 | 0.001 | 0.001 | 0.0 | 0.0 |
| Has inference | 0.0 | 0.0 | 0.006 | 0.08 | 0.002 | 0.002 | 0.263 | 0.263 |
| Has composition | 0.591 | 0.594 | 0.571 | 0.957 | 0.501 | 0.507 | 0.907 | 0.907 |

| Relation | Compound-causes-Side Effect | | Gene-regulates-Gene | | Gene-interacts-Gene | | Compound-downregulates-Gene | |
|---|---|---|---|---|---|---|---|---|
| Dataset | Hetionet | PhMBNet | Hetionet | PhMBNet | Hetionet | PhMBNet | Hetionet | PhMBNet |
| Relation triples | 138944 | 154511 | 265672 | 265667 | 147164 | 147133 | 21102 | 231156 |
| Matching rate | 0.909 | 0.817 | 0.999 | 1.0 | 0.999 | 1.0 | 0.997 | 0.098 |
| Test triples | 12630 | 12630 | 26566 | 26566 | 14713 | 14713 | 2105 | 2105 |
| Unique heads | 1071 | 1358 | 4634 | 4634 | 9526 | 9525 | 734 | 2631 |
| Unique tails | 5701 | 6023 | 7048 | 7047 | 14084 | 14073 | 2880 | 21912 |
| Head out-degree | 245 | 2186 | 203 | 254 | 214 | 267 | 515 | 4144 |
| Head out-degree s.r. | 201 | 182 | 104 | 104 | 54 | 54 | 225 | 1413 |
| Unique out-relations | 5 | 19 | 6 | 10 | 5 | 10 | 5 | 20 |
| Tail in-degree | 164 | 254 | 309 | 370 | 106 | 145 | 252 | 116 |
| Tail in-degree s.r. | 164 | 193 | 208 | 208 | 27 | 27 | 20 | 15 |
| Unique in-relations | 1 | 3 | 8 | 12 | 7 | 11 | 8 | 11 |
| Has inverse | 0.0 | 0.0 | 0.003 | 0.003 | 0.006 | 0.006 | 0.0 | 0.002 |
| Has inference | 0.0 | 0.142 | 0.006 | 0.003 | 0.006 | 0.006 | 0.001 | 0.164 |
| Has composition | 0.366 | 0.893 | 0.881 | 0.886 | 0.703 | 0.713 | 0.913 | 0.800 |

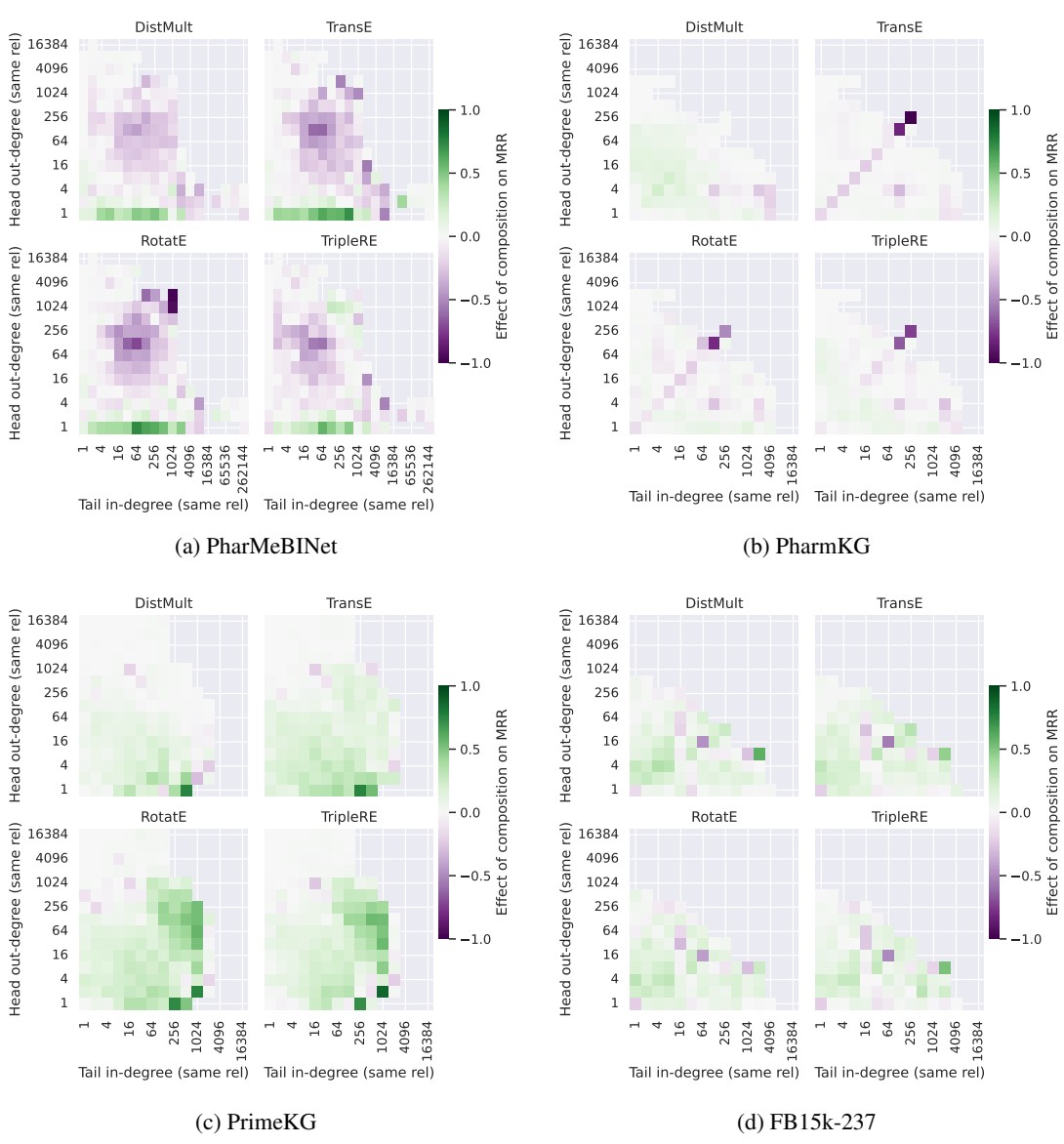

(a) PharMeBINet

(b) PharmKG

(c) PrimeKG

(d) FB15k-237

Figure C.6: The effect of having compositions on MRR, triples grouped by their head and tail degrees.

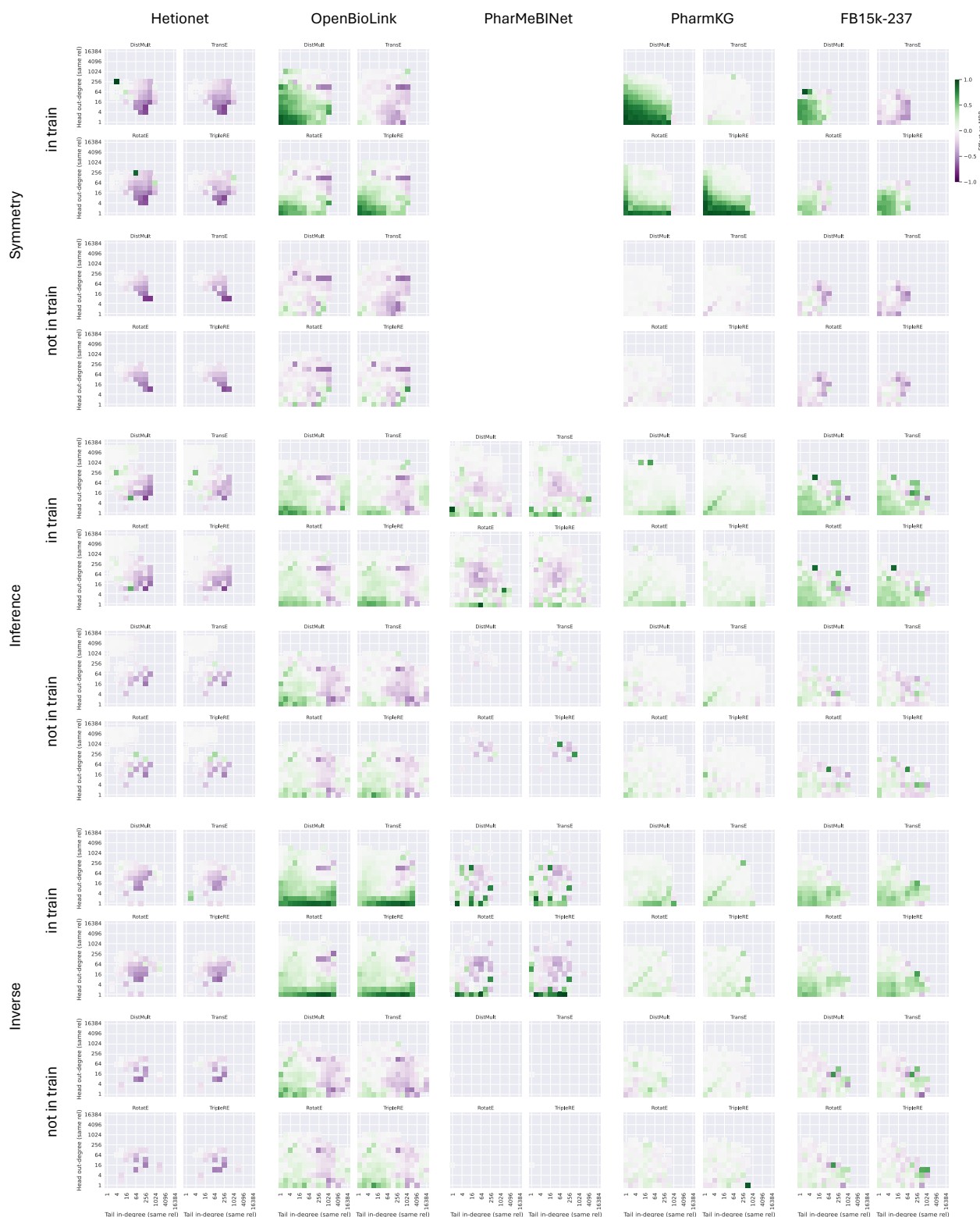

Figure C.7: The effect on MRR of being symmetric and having inference/inverse, distinguishing based on whether the counterpart edge is present or absent in in the training data. Triples grouped by their head and tail degrees.

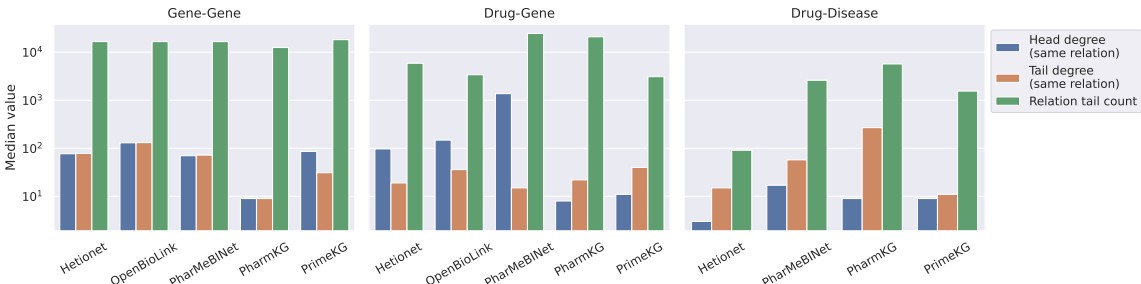

(a) Median head out-degree and tail in-degree of same relation and number of unique relation tail entities.

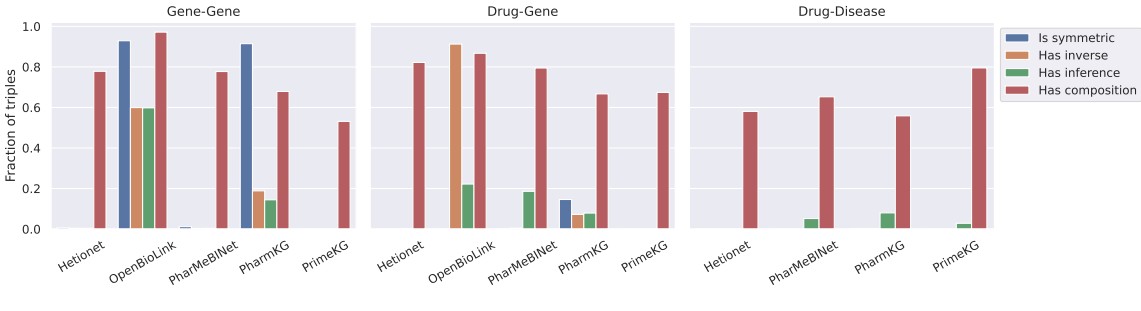

(b) Frequency of edge patterns.

Figure C.8: Statistics of topological properties for the interaction types investigated in Section 4.2.

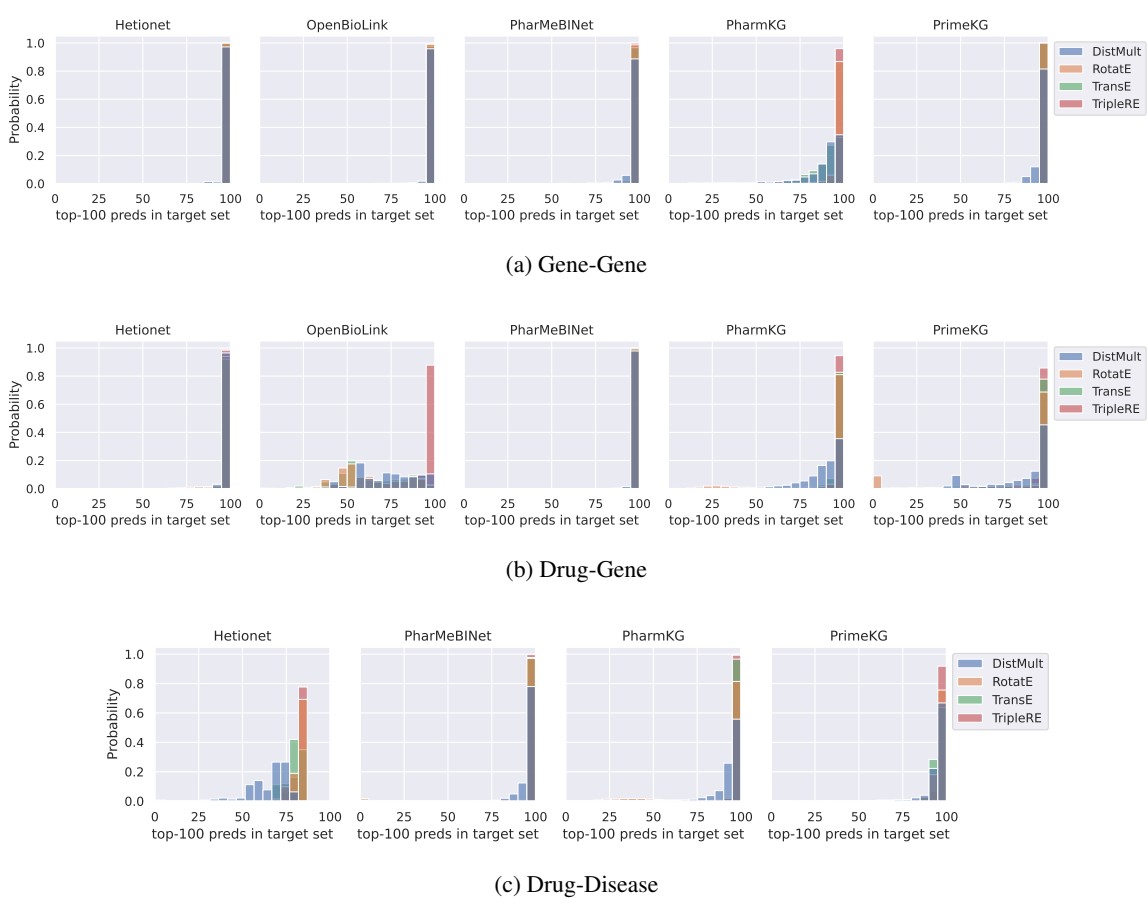

Figure C.9: Demixing for the interaction types investigated in Section 4.2. For each test query, we compute how many of the top-100 predictions made by the model are contained in the set of entities used as tails by triples of the considered relations.

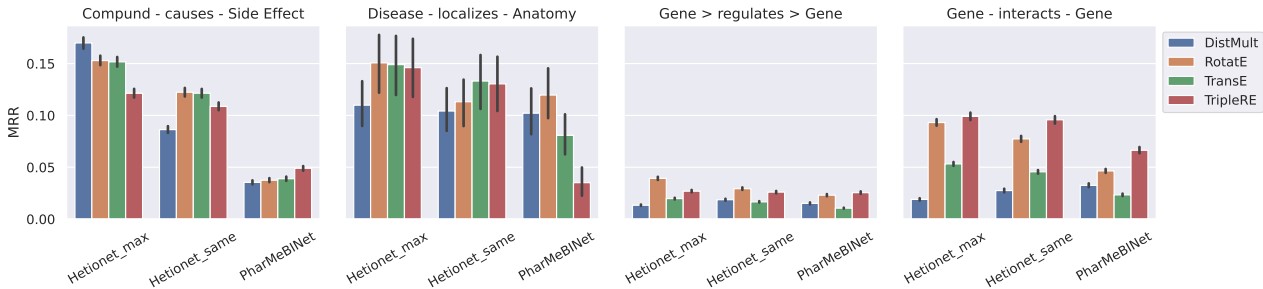

Figure C.10: MRR comparison for additional relation types, when testing on a set of common edges between Hetionet and PharMeBINet.