# OpenReview forum: "Towards Linking Graph Topology to Model Performance for Biomedical Knowledge Graph Completion"
_ICML.cc/2024/Workshop/ML4LMS — ML4LMS Poster_

### Official Review · Reviewer_EqPh · 2024-05-29

**Rating:** 6
**Confidence:** 4

**Review:**

Summary

The paper investigates the topological properties of biomedical knowledge graphs (KGs) and their impact on the downstream task of knowledge graph completion. The work explores topological properties: edge cardinalities and relational patterns in six public biomedical KGs and examines how these properties influence model performance. The authors commit to releasing all models, results, and software associated with the research. While previous studies have explored the impact of relational patterns on validation metrics across different train/test splits, this paper uniquely focuses on the specific topological characteristics of biomedical KGs and their implications on widely used evaluation metrics such as MRR.

Strengths

- The paper looks at graph properties: edge cardinalities and relational patterns such as symmetry, inversion, and composition within the biomedical KGs. These properties are analyzed to understand their relevance and impact on the performance of KG completion models.

- Extensive experiments were conducted using four selected KG embedding (KGE) models. The rationale for choosing these models should be clarified, given the availability of many models in the KGE literature.

 - The models were evaluated on six public biomedical KGs to report the effect of different topological properties on model performance.

- The results demonstrate that different topological properties significantly affect the performance of KGE models. For instance, the impact of edge cardinality and specific relational patterns on predictive accuracy is highlighted through detailed experiments.

- The case study in section 4.3 provides examples from PharMeBINet, demonstrating the model's performance on different relation types such as "downregulates," "binds," "expresses," and "covaries." Although, an interesting visualization, a more thorough discussion of these properties and their implications would improve the readability of the paper. For instance, what relation type (symmetry, composition, etc) do they follow?


 Points that could perhaps be useful to improve the paper:

- Section 2 would benefit from including real-world examples from biomedical KGs to illustrate the relevance of different topological properties. Perhaps provide an example. Consider three nodes A: Cardiovascular disease, B: Coronary Artery Disease and C: Acute Myocardial Infarction. Now assume there is a relation "subtype_of". In KG there can be triples (B, “subtype_of”, A), (C, "subtype_of", B). Now due to transitivity, one could infer (C, “subtype_of”, A). Likewise, consider examples of other types of relations that might be present in chosen KGs. Having such examples in motivation and introduction will improve the readability of the paper.

-  The choice of the four selected KGE models lacks clear motivation. It is not apparent why not run evaluation across more commonly used models. Ideally, one would consider a different combination of models that can capture either S, INV, INV or C relation or all. However, the chosen model in A.3 only transE doesn’t capture S and DistMult doesn’t capture INV or C. If INF is captured by all four models is it a reasonable one to consider?

- Conclusion is too verbose. Perhaps summarize it in a short paragraph and discuss the possible implication of capturing these relations in a clinical setting for example drug discovery.

- Table A.2: Improve clarity by adding a column for the total number of triples in all KGs. Although provided in Table A.1 it would be good to have it here as well for better readability. Discuss the properties of relations in at least one dataset in a bit more detail, along with edge cardinality and model performance.

- Figure C.1: Clarify the correspondence of different datasets to the plots. There are four plots, one for each relation and each plot has four curved ones for each method. I don’t see what corresponds to different datasets.

- PrimeKG has 0 symmetry, 0 inverse and almost negligible inference triples. Is that a property of the database, what types of relations are present there some discussion would be useful.

- Are there relations beyond the four outlined in Table A.2. As these are fractions from total triples should they sum to 1? Can the same triple satisfy multiple properties? Perhaps a more clear discussion would benefit the paper.


- Overall a general suggestion: the caption of the table and figures should be self-explanatory for a reader.  Line 36 reference to Bonner et al 2022a. Reference to other related work on biomedical KG is missing. It is okay to reference a review paper when the field is exhausted with papers. However, it is important to cite relevant papers where work was done.


Overall I think it is a good paper that takes the initiative to investigate the implication of various graph properties of KGE models on KGC for real-world KGs. Readability can be improved by taking examples to discuss and motivate the work.

---

### Official Review · Reviewer_mFGe · 2024-06-05

**Rating:** 7
**Confidence:** 3

**Review:**

This paper investigates the performance of different models on Biomedical Knowledge Graph Completion tasks, which is of empirical interesting in many cross-modal biological applications. The authors conduct a series of experiments to analyze the correlation between model performance and topological properties, which can encourage a wider discussion on the KG completion tasks in biological community.

---

### Official Review · Reviewer_C1du · 2024-06-08

**Rating:** 7
**Confidence:** 3

**Review:**

This paper mainly studied the relationship between biomedical Knowledge graph(KG) topology and the prediction accuracy of a variety of KGE models. The findings, for example edge topological patterns also impact predictive accuracy, can be interesting and relevant. Overall, the experiments are solid and paper is well written. Comprehensive experiments across 6 public available KGs and 4 KGE models can support the claims.